Resource

# Performance and robustness analysis reveals phenotypic trade-offs in yeast

Cecilia Trivellin[1], Peter Rugbjerg[1,2], Lisbeth Olsson[1]

**To design strains that can function efficiently in complex industrial settings, it is crucial to consider their robustness, that is, the stability of their performance when faced with perturbations. In the present study, we cultivated 24 *Saccharomyces cerevisiae* strains under conditions that simulated perturbations encountered during lignocellulosic bioethanol production, and assessed the performance and robustness of multiple phenotypes simultaneously. The observed negative correlations confirmed a trade-off between performance and robustness of ethanol yield, biomass yield, and cell dry weight. Conversely, the specific growth rate performance positively correlated with the robustness, presumably because of evolutionary selection for robust, fast-growing cells. The Ethanol Red strain exhibited both high performance and robustness, making it a good candidate for bioproduction in the tested perturbation space. Our results experimentally map the robustness–performance trade-offs, previously demonstrated mainly by single-phenotype and computational studies.**

## Introduction

To achieve high yields while ensuring reproducibility and high-quality in bioprocesses, it is essential to address the factors contributing to process variability. The variable conditions, also referred as perturbations, can be triggered by the bioprocess environment (temperature, nutrient gradients, changes in raw materials), intracellular factors (noisy gene expression, genetic mutations) (Olsson et al, 2022), and different bioprocess steps (cell propagation, fermentation) (Tomás-Pejó & Olsson, 2015; Keil et al, 2019). The set of all perturbations present in the bioprocess is referred to as the perturbation space, and can be both predictable and stochastic. Generally, the bioprocess perturbation space has a negative impact on strain performance compared with its ideal and well-controlled lab-scale space, both in terms of specific growth rates and production of valuable metabolites. Therefore, industrial strains must exhibit consistent performance, that is, robustness, to avoid loss of product quantity and quality or larger costs (Huang et al, 2016; Mohedano et al, 2022).

Microbial robustness refers to the ability of cells to maintain a stable performance when exposed to a perturbation space (Kitano, 2004; Masel & Siegal, 2009; Levy et al, 2012; Félix & Barkoulas, 2015; Olsson et al, 2022). Microbial robustness is typically assessed for phenotypes related to industrial performance (titer, rate, and yield), but it can be extended to cell volume, cell viability, gene expression or indirect measurements of cellular parameters such as fluorescence. For simplicity, microbial robustness is referred hereafter as robustness.

Numerical assessment of robustness can be used to quantify performance stability (Steensels et al, 2014; da Conceição et al, 2015; Louis, 2016; Mestek Boukhibar & Barkoulas, 2016). We have previously proposed and validated a high-throughput methodology to quantify robustness in multiple phenotypes, resulting in a dimensionless negative number, where the theoretical zero represents a completely robust, non-changing phenotype (Trivellin et al, 2022). Our methodology is built on a subset of phenotypes (cellular functions) that can be measured experimentally. Using a series of single experimental perturbations designed to simulate bioprocess conditions, robustness measures the variation of the performance of interest with respect to its average across multiple perturbations. In addition, robustness quantification allows the exploration of a broad range of physiological phenomena (e.g., trade-offs), which would otherwise be challenging to investigate using standard methods (e.g., fermentation profiling in bioreactors).

Microorganisms appear to exhibit trade-offs between performance and robustness of one or more phenotypes (Kitano, 2007, 2010), which could be important for improving strain and process development. For example, in bacteria, antibiotic resistance and extracellular enzyme production have been shown to trade-off with the specific growth rate and a similar trade-off has been observed between melanin production and specific growth rate in fungi (Andersson, 2006; Ramin & Allison, 2019; Lovero & Treseder, 2021). To the best of our knowledge, trade-offs between performance and robustness have been investigated mostly in single-phenotype and in silico studies (Ibarra et al, 2002; Stelling et al, 2002; Fischer & Sauer, 2005). Experimental validation of trade-offs could explain why strains optimized for maximum performance are less capable of coping with environmental stresses and fluctuations (i.e., present lower robustness).

[1]Department of Life Sciences, Division of Industrial Biotechnology, Chalmers University of Technology, Gothenburg, Sweden    [2]Enduro Genetics ApS, Copenhagen, Denmark

Correspondence: lisbeth.olsson@chalmers.se

In the present study, we applied our previously developed robustness quantification method (Trivellin et al, 2022) to a large dataset of yeast responses to perturbations. The dataset contained more than 10,000 phenotypic data points obtained upon cultivation of 24 *Saccharomyces cerevisiae* strains under 29 different growth conditions simulating bioethanol production from second-generation biomass (perturbation space). A culture transfer step was included as additional perturbation to explore the method's versatility in assessing robustness within different bioprocess steps.

The combination of exploratory data analysis on the large dataset with quantification of robustness allowed us to map perturbation-specific influences on performance and robustness, and identify strains with robust phenotypes. We proved that exposing a cell culture to a perturbation during pre-cultivation, significantly increased the performance associated with the specific growth rate during subsequent cultivation. Finally, correlation tests revealed trade-offs between robustness and performance (measured as biomass and ethanol yield and cell dry weight.) Although our earlier work primarily focused on the development and validation of the robustness quantification method itself, in the current study our aim was to demonstrate its practical application. Specifically, our study provides a model for integrating performance and robustness data to uncover phenotypic trade-offs in yeast, which is a critical aspect of strain engineering. Our findings demonstrate that strongly performing cells under one condition may be less robust in others, underscoring the importance of considering both factors in the design process.

# Results

### High-throughput characterization reveals how perturbations in medium components influence the phenotypes of *S. cerevisiae* strains

A total of 24 *S. cerevisiae* strains were examined in the present study. They included two well-characterized laboratory strains, four industrial strains employed in bioethanol production or baking, and 18 LBCM strains isolated from cachaça fermentation plants (see the Material and Methods section) (De Araújo Vicente et al, 2006). Each strain was cultivated in microtiter plates containing chemically defined Delft medium plus a single component simulating perturbations found during industrial lignocellulosic bioethanol fermentations, such as acetic acid released during the hydrolysis of hemicellulose (Jönsson & Martín, 2016). The single conditions were grouped according to similarity between the physiological responses they elicited in yeast. Specifically, the acids (lactic, levulinic, acetic, and formic acid) were grouped together, as were the pentoses (xylose and arabinose) and hexoses (galactose, glucose, and mannose). The aldehydes, including vanillin, 5-hydroxymethylfurfural, and furfural, were classified together, whereas NaCl and ethanol were considered single components. For each cultivation, five phenotypes (specific growth rate, lag phase, final cell dry weight, biomass yield, and ethanol yield) were calculated for a total of 10,295 data points.

Although it has been reported that industrial strains grow faster than laboratory strains (Kong et al, 2018; Yi & Alper, 2022) because of their ability to handle perturbations in industrial settings, no significant differences were detected between the two groups. Overall, Ethanol Red showed the highest mean performance with respect to all phenotypes except the ethanol yield, in which case PE2 attained the highest mean value (Fig 1A). The ethanol yield of the PE2 strain was more than double the average calculated across all strains. The PE2 strain has been reported to have a high $CO_2$ production, final ethanol production (close to 95% of the theoretical yield in *E. globulus* wood hydrothermal hydrolysate anaerobic fermentation), fast sugar utilization and faster degradation of furfural and 5-hydroxymethyl-2-furaldehyde (Pereira et al, 2010, 2014; Soares-Costa et al, 2014). Furthermore, PE2 has also been shown to dominate and persist in Brazilian distilleries probably because of its high viability and high specific growth rate (Raghavendran et al, 2017; Araújo et al, 2018). In our study, the specific growth rate of PE2 was 20% lower than the average across all strains.

To assess for correlations among phenotype performances, we carried out Spearman correlation tests. The maximum specific growth rate correlated negatively with lag phase ($P < 2.2 \times 10^{-16}$) (Fig S1), confirming previous observations (Basan et al, 2020). Instead, positive significant correlations were observed between specific growth rate and ethanol yield, biomass yield, and cell dry weight. However, when splitting the data into groups of conditions, correlations became nonsignificant in the case of hexoses, both for ethanol and biomass yield, and in the case of pentoses and NaCl for ethanol yield. The Spearman tests revealed a positive monotonic relationship between biomass yield and end-of-cultivation cell dry weight with respect to specific growth rate. Negative correlations were observed in single strains, such as Ethanol Red and PE2; although no overall negative relationship between production (calculated as ethanol yield) and growth (maximum specific growth rate) was detected in this perturbation space.

Analysis of strain performance revealed a substantial negative impact of groups of conditions on the different phenotypes, whereas no such effect was observed in growth medium containing only hexoses. The only exceptions were biomass yield in the presence of pentoses or hexoses, and maximum specific growth rate in the presence of hexoses or acids (Wilcoxon test, $P$ = ns) (Fig 1B). The negative effect of aldehydes, acids, ethanol, and NaCl on the phenotypes confirmed earlier studies (Adeboye et al, 2014; Tekarslan-Sahin et al, 2018; Caspeta et al, 2019; Guaragnella & Bettiga, 2021). Ethanol Red, PE2, and the LBCM strains are highly tolerant towards ethanol (Demeke et al, 2013) and lignocellulosic inhibitors (Wallace-Salinas & Gorwa-Grauslund, 2013; Araújo et al, 2018; Cunha et al, 2019). Here, Ethanol Red and some LBCM strains displayed higher tolerance (higher specific growth rate) towards aldehydes (Fig 1A).

The presence of acids lowered the yields for all strains; although no significant difference in specific growth rate was observed between medium containing acid and not (Fig 1B). Considering fermentation kinetics, a decreased yield of biomass on substrate while keeping the specific growth rate constant may suggest an increase in the specific rate of substrate consumption. Weak acids are not inhibitory enough to slow down the anabolism; therefore

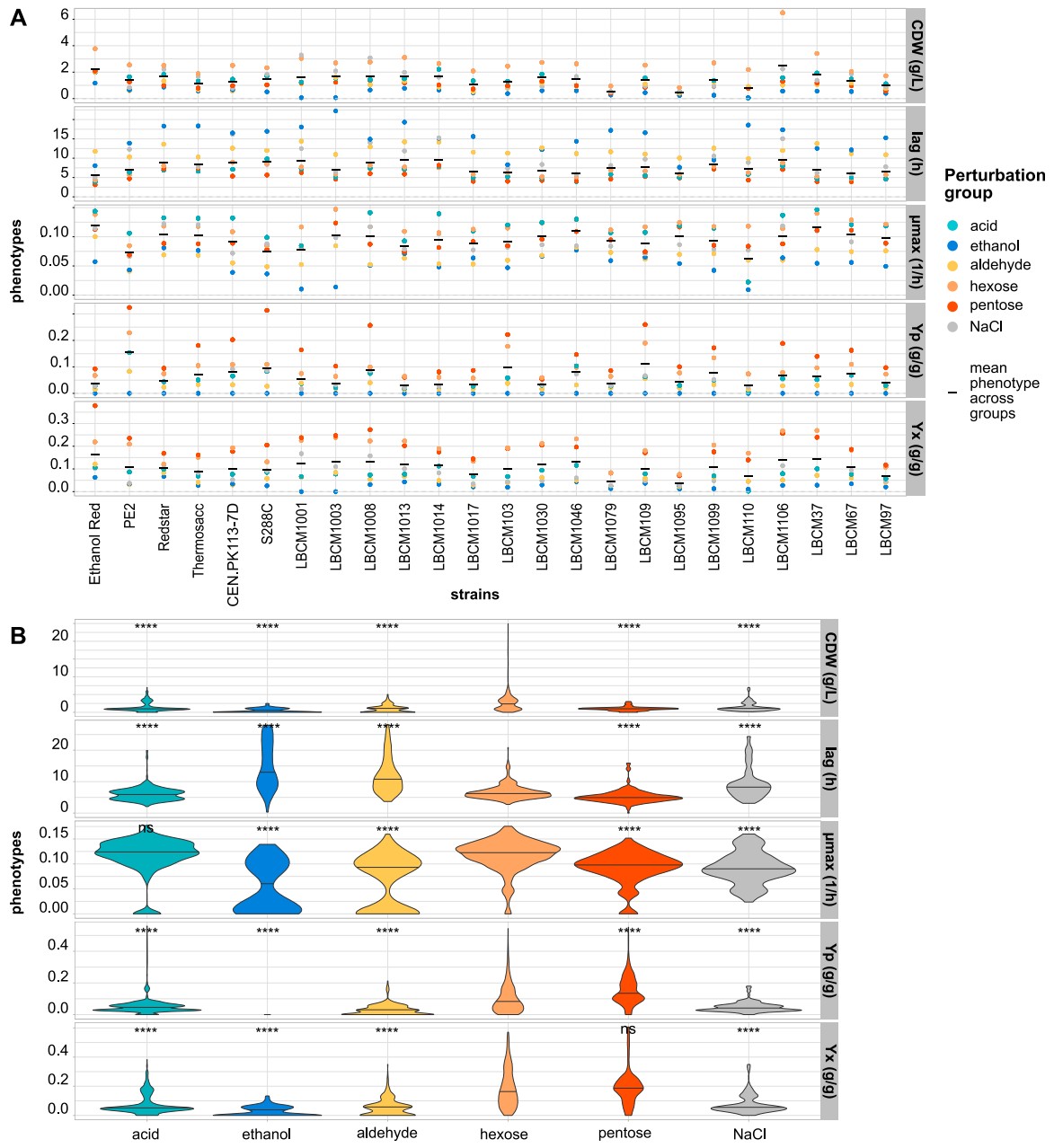

**Figure 1. Yeast phenotypes are impacted significantly by environmental perturbations.**
**(A)** Phenotypic data averaged across groups of perturbations for five phenotypes ($\mu$max, maximum specific growth rate; Yp, ethanol yield; Yx, biomass yield; CDW, cell dry weight; and lag phase) and 24 *S. cerevisiae* strains (X-axis). **(B)** Data distributions of the five measured phenotypes for each group of perturbations and 24 strains (X-axis). Data information: in panel (A), each colored dot corresponds to the mean across triplicates and groups of perturbations. In panel (B), the black line inside the colored area represents the median of the distribution. Differences between groups and hexoses were assessed with a Wilcox test (n = 10,295; ns, not significant; ****$P < 1.4 \times 10^{-5}$).

growth can proceed at high specific rate. However, ATP is required for counteracting the effects of the acidification, which diverts the carbon source from anabolism to catabolism. This decreases the yield and increases the specific substrate consumption rate, and the specific production rate of energy-related primary metabolites. In the presence of up to 7 g/liter lactic acid, all phenotypes displayed higher or comparable performance to the control containing 20 g/liter glucose. This was likely because the pKa of lactic acid is 3.79, and at a pH of 5, most of the acid existed in its dissociated

form, making it less likely to penetrate the cells. Lactic acid is encountered during bioethanol production because of contamination with *Lactobacillus* spp. Interestingly, *Lactobacillus amylovorus* has been shown to be beneficial (3% higher ethanol yields) or neutral to yeast fermentation (Senne de Oliveira Lino et al, 2021).

A higher average ethanol yield was noted across all strains when comparing pentoses and hexoses (54% higher, *P* < 0.001). S288C and LBCM1008 showed a significantly higher ethanol yield in medium containing 5 g/liter glucose plus various amounts of xylose or

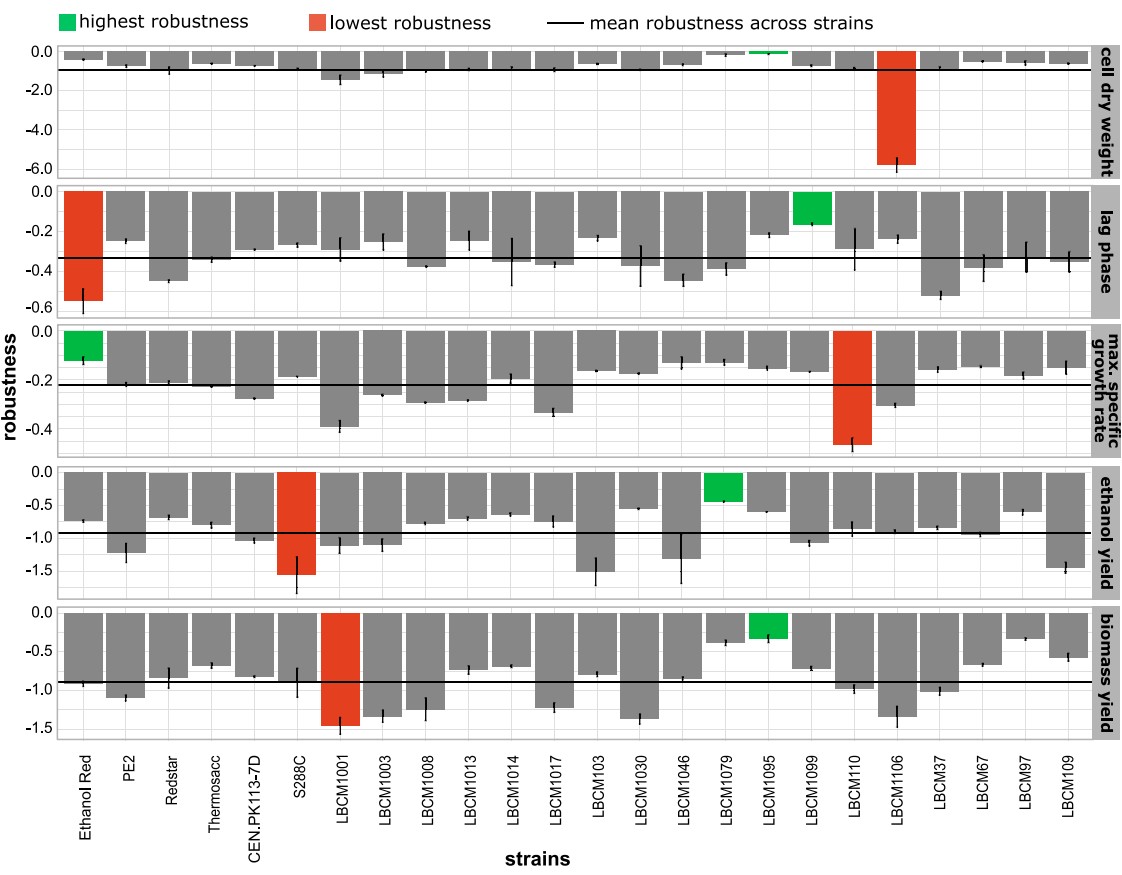

**Figure 2. Robustness quantification reveals significant differences between strains for five phenotypes.**
Data information: the five horizontal plots correspond to the five phenotypes tested in the study. Strains are shown on the x-axis. Each robustness value was calculated with Equation (1). Error bars denote the standard error of the mean (n = 3).

arabinose compared with medium containing only 20 g/liter glucose (Fig 1A). The other phenotypes did not exhibit an increment in the presence of pentoses (Fig 1B).

Owing to the meticulous evaluations of five different phenotypes in 24 strains, we were able to point out the best performing strains and illustrate how their performance was influenced by different inhibitors with a fast and comprehensive method in small scale.

### Ethanol Red is a unique compromise for high robustness and performance

In the present study, we applied our previously developed methodology to quantify the robustness of five phenotypes in 24 strains within a set perturbation space (Equation (1), see the Materials and Methods section) (Trivellin et al, 2022).

Robustness varied according to phenotype and strain (Fig 2). LBCM1079 and LBCM1095 displayed constantly higher robustness compared with other strains; whereas LBCM1001 and LBCM1106 presented overall low robustness values. Red Star exhibited strong robustness for some phenotypes, but low robustness for others. LBCM1106 exhibited a significantly lower robustness for the cell dry weight mainly attributed to its outstanding performance in the presence of hexoses (Fig 1A). When considering all strains,

robustness was generally higher for lag phase and maximum specific growth rate (−0.2 and −0.3, respectively) than for other phenotypes, such as ethanol yield (−0.9), biomass yield (−0.9), and cell dry weight (−0.9). Larger experimental variation in end-of-cultivation cell dry weight and yields contributed to the lower robustness.

LBCM1095, LBCM1079, and LBCM97 showed the highest mean robustness: −0.3, −0.3, and −0.4, respectively. LBCM97 has been described as highly tolerant towards ethanol, aluminum, and a broad pH range (da Conceição et al, 2015; van Dijk et al, 2020). The mean robustness of Ethanol Red was −0.5, mostly because of low specific growth rate in the presence of ethanol. Ethanol Red is very robust when fermenting sugars from grain mash at a high temperature (Wallace-Salinas & Gorwa-Grauslund, 2013), and highly performing in fed-batch fermentation on molasses (Demeke et al, 2013). Owing to the way robustness is quantified (Félix & Barkoulas, 2015; Trivellin et al, 2022), a poorly performing phenotype would result in elevated robustness values if it behaved consistently across multiple conditions. To fully understand how a strain responds to various perturbations, a comprehensive overview of both its performance and robustness is required. When evaluating robustness and performance simultaneously, Ethanol Red stood out as the best compromise. This strain displayed an outstanding performance in four out of five phenotypes, and ranked among the highest for robustness. Therefore, even though the perturbation set

chosen for the study was not specific to starch substrates, which are preferred by Ethanol Red (Cripwell et al, 2019), the genetic and physiological make-up of this strain allows it to be highly performing even in perturbation spaces other than those associated with starch fermentation.

## The effect of groups of conditions on robustness helps identify key factors that significantly influence strain performance

The overall performance of some strains (e.g., LBCM37) varied greatly within the perturbation space, which led to low robustness compared to other strains (Fig 1A, spread of the data points for each strain). To better understand such variation, we tested the contribution of each group of conditions (pentoses, hexoses, NaCl, ethanol, aldehydes, and acids) to the robustness of each phenotype. The robustness calculated with Equation (1) was compared with the robustness obtained with the same equation, but excluding each group of conditions (Fig S2).

Overall, pentoses, acids, and NaCl had a neutral impact on the robustness, except for the robustness of the lag phase which was negatively impacted by the pentoses and the robustness of the biomass yield, negatively impacted by both NaCl and acids. Ethanol and aldehydes had a negative impact on the robustness of all phenotypes, except for some of the strains whose lag phase robustness was not affected (for example, the laboratory strains). Ethanol Red and LBCM110 were the only two strains, whose cell dry weight was not affected by aldehydes.

Calculating the influence of single groups of conditions on the robustness of a specific strain serves three purposes. First, it reveals which conditions cause the largest spread in the distribution of data (Fig 1A). Second, it suggests which conditions should be included in the tested perturbation space. Assuming all relevant stochastic and predictable perturbations (from an extracellular or intracellular environment) are included in the perturbation space, testing the influence of certain groups of conditions on robustness could reduce the number of conditions to assess. The number of tested perturbations should nevertheless be statistically significant for the robustness value to have a meaning. Third, if the robustness of a specific phenotype is not affected only in a few strains, the latter could reveal metabolic mechanisms responsible for such observation.

## Negative correlations between performance and robustness confirm presumed trade-offs

Trade-offs between robustness and performance have been hypothesized previously (Kitano, 2007, 2010; Quinton-Tulloch et al, 2013). For example, in-silico studies have suggested that cells investing resources in anticipation of environmental changes display suboptimal growth (Fischer & Sauer, 2005). To determine the trade-offs between performance and robustness, a three-dimensional matrix of strains, perturbations, and phenotypes was created. Spearman's correlation tests were carried out on robustness and performance to display monotonic relationships among the measured values (Fig 3). Negative correlations can provide evidence of phenotypic trade-offs, if the observed phenotypic response is directly caused by the environmental stimulus received

(Fox & Stevens, 1991). Because complex genetic architecture and regulatory networks determine a specific phenotype, generalization is not possible. Here, we observed potential trade-offs connected to the applied perturbation space.

Negative correlations (–0.8 to –0.6) were identified between robustness and performance of cell dry weight, biomass yield, and ethanol yield ($P < 0.05$) (Fig 3A). This result suggested a performance–robustness trade-off, whereby low performances were associated with high robustness and vice versa. The only positive correlation between performance and robustness was observed for maximum specific growth rate (Fig 3B). Instead, no significant correlation was found for the lag phase.

The estimated standard error for the Spearman correlation between performance and robustness was around 0.1 (see the Materials and Methods section) for ethanol yield, biomass yield, and cell dry weight, and around 0.2 for specific growth rate. The error increases with small sample size and weaker ($R < 0.6$) correlation coefficients. Some strains displayed more evident performance-robustness trade-offs, but only for certain phenotypes, for example Ethanol Red's lag phase had the highest performance but the lowest robustness among all the strains (Fig S3). Ethanol Red low lag phase robustness is attributed to its ability to grow, after a very long lag phase, in harsh conditions.

Positive significant correlations were observed between robustness of biomass yield and maximum specific growth rate with the robustness of cell dry weight (0.8 and 0.7 respectively), and between the robustness of biomass yield with the robustness of maximum specific growth rate (0.5) (Fig 3A). Significantly negative correlations were observed instead between performance and robustness of various pairs of phenotypes, including robustness of cell dry weight and lag phase or ethanol yield, robustness of biomass yield and cell dry weight, robustness of lag phase and specific growth rate, and robustness of specific growth rate and lag phase.

Our results point to the phenotype-specific behavior of robustness (Fig 2), thereby supporting previous evidence (Barkai & Leibler, 1997; Carlson & Doyle, 2000; Félix & Barkoulas, 2015; Trivellin et al, 2022). Even though robustness cannot be considered a general property of a system, the positive correlations observed among robustness values (Fig 3A) may suggest that biomass yield, specific growth rate, and cell dry weight are regulated and stabilized by similar or shared mechanism.

In summary, the observed correlations validated the hypothesized trade-offs between robustness and performance for cell dry weight, biomass yield, and ethanol yield. Moreover, they provide a powerful tool for further investigations of possible trade-offs.

## 95% of the tested strains showed an increased maximum specific growth rate upon transfer to the same medium

When designing the perturbation space, it is crucial to consider all the bioprocess steps, including preculture or inoculation, as they can have a negative impact on the process outcome. In the present study, we expanded the perturbation space by including a culture transfer and investigated its impact on the specific growth rate of each strain. We compared the maximum specific growth rate in the first cultivation step, with the one in the second cultivation after

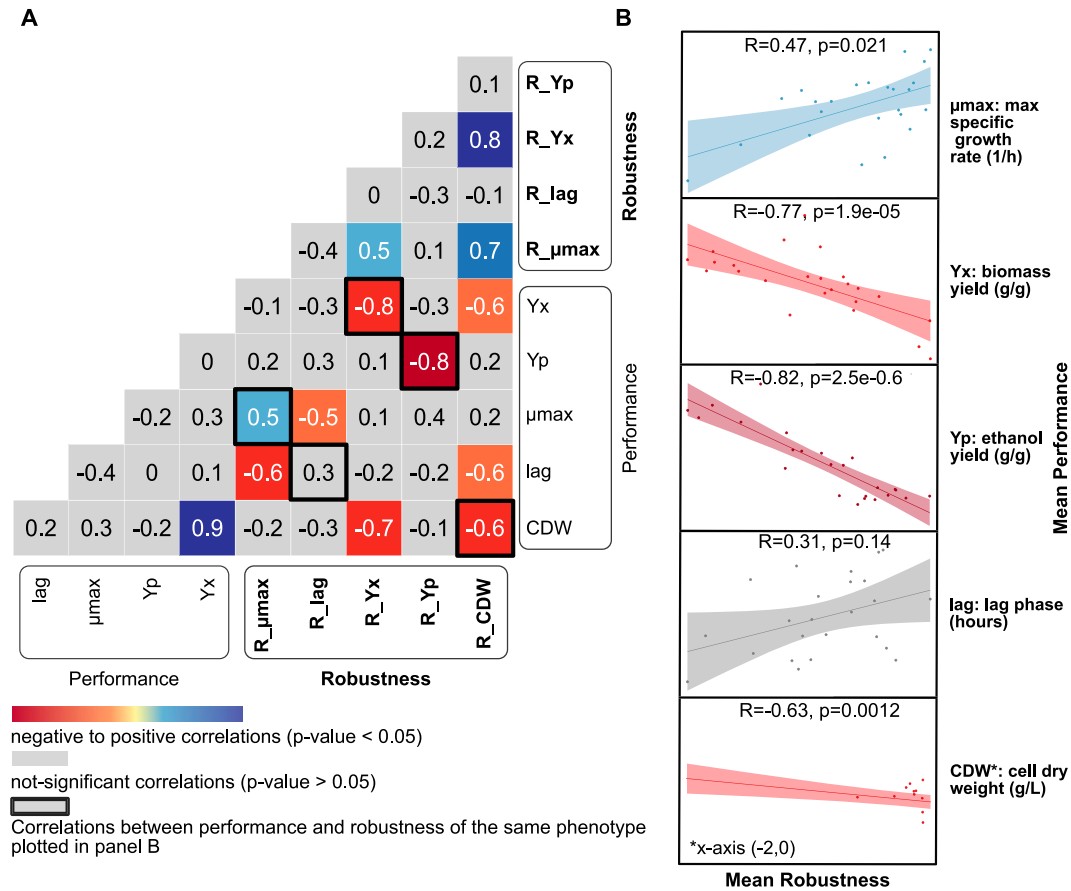

**Figure 3. Spearman correlations between the performance and robustness of five different phenotypes show potential performance–robustness trade-offs.**
**(A)** Spearman correlation matrix between performance and robustness for five phenotypes. **(B)** Correlations between mean performance and mean robustness of the five measured phenotypes. Data information: in panel (B), each dot corresponds to a single strain. A linear regression line is plotted in each panel mainly to visualize the direction and strenght of the relationship. R = Spearman correlation coefficient; *P* = *P*-value; n = 24.

transfer (see the Materials and Methods section). We calculated the improvement in maximum specific growth rate in terms of performance (Equation (2)). A strong and significant positive correlation between the first and second maximum specific growth rates was found (Fig 4A). No notable discrepancies among the groups were detected when dividing the correlations into various perturbation categories. The correlation coefficients ranged from 0.6 for ethanol to 0.8 for NaCl. The *P*-value was > $2.2 \times 10^{-16}$ for all groups of conditions, confirming statistically strong correlations. Sugars exhibited a slightly lower correlation coefficient compared with other conditions probably because of a weaker influence of sugars on the pre-cultivation compared with harsher conditions (e.g., aldehydes). Fourteen out of 24 strains presented a minimum 20% increase in the specific growth rate of the second cultivation, with nine strains boosting performance by more than 50%. The only strain that did not show mean improvement was S288C (Fig 4B).

The improved maximum specific growth rate in the second cultivation was strongly influenced by the condition and differed for each strain. LBCM1001, LBCM1017, and LBCM110 displayed outstanding improvement in specific growth in the presence of ethanol and, in the case of LBCM1017 and LBCM110, also in the presence of acids. CEN.PK113-7D was the only strain that never exhibit a

decrease in its maximum specific growth rate during the second cultivation (Fig 4B). Interestingly, the higher was the concentration of certain inhibitors (acids, furfural, 5-hydroxymethylfurfural and mannose) in the medium, the larger was the percentage of improvement of the maximum specific growth rate (Fig S4). A correlation between the percentage of improvement of the maximum specific growth rate (%P) and the robustness of each of the phenotypes was not found. Overall, exposing a strain to the same stressor prior main cultivation had a positive effect on its tolerance (increased specific growth rate).

## Discussion

In the present study, we investigated a large volume of phenotypic data from 24 *S. cerevisiae* strains grown in 29 different conditions, simulating the lignocellulose biomass fermentation perturbation space. Phenotypic characterization combined with robustness quantification allowed us to map the condition-specific influence on performance and robustness. Accordingly, we demonstrated the positive influence of the propagation step on performance. We also discovered and proved experimentally the trade-off between

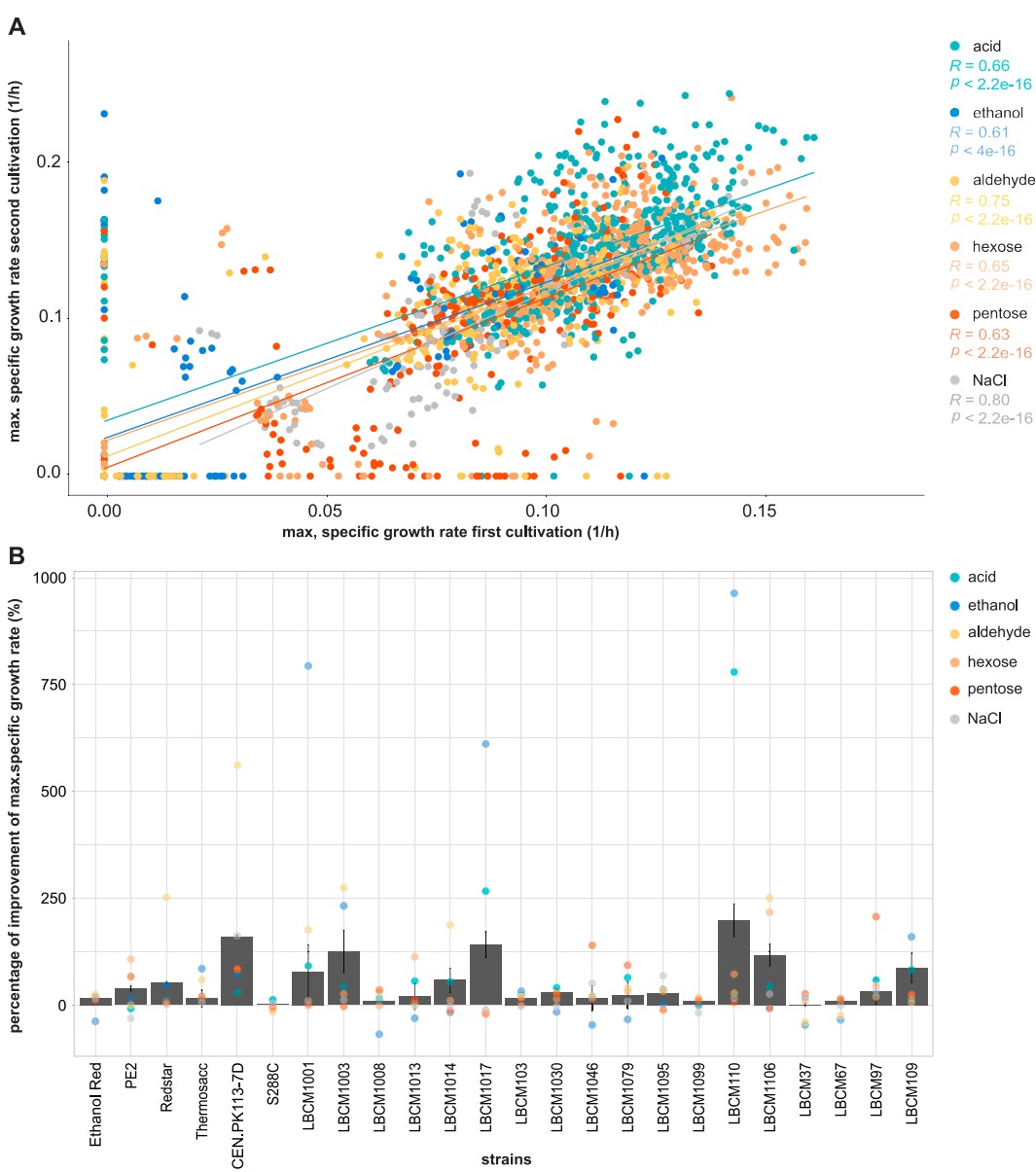

**Figure 4.   Exposure to the same media improves the specific growth rate performance of tested strains during cultivation.**
**(A)** Plot of the specific growth rate of the first cultivation and second cultivation for each strain and perturbation. **(B)** Percentage of improvement of the maximum specific growth rate calculated using Equation (2). Data information: in panel (A), each color represents a different group of perturbations and includes the corresponding linear regression line (mainly to check direction and strenght of the relationship). Spearman correlation coefficients are shown in the right part of the graph for each group of perturbations, together with the *P*-value denoting the significance of the correlation. *P* < 0.005 indicates strong statistical significance. In panel (B), the mean improvement in performance of the maximum specific growth rate (%P) was calculated with Equation (2) across perturbations for each strain (barplots). The SD is shown for each strain calculated from the triplicates. The colored dots correspond to the %P divided by group of perturbations for each strain.

performance and robustness with respect to biomass yield, ethanol yield, and cell dry weight.

Each group of conditions affected performance in a consistent manner. Similar phenotypic responses may be linked to shared metabolic processes, such as the accumulation of reactive oxygen species under acid and aldehyde stress (Allen et al, 2010; Guaragnella & Bettiga, 2021). Using biosensors to examine the intracellular state can be a valuable approach for studying these patterns (Torello Pianale et al, 2022). By exploring the full range of phenotypes within the perturbation space, we were able to identify novel behaviors. For

example, the presence of pentoses and low glucose (5 g/liter) led to an increase in ethanol yield, despite pentoses not being used as carbon and energy source by *S. cerevisiae*. We hypothesize that this might be attributed to an increased efficiency of some hexose transporters (specifically Hxt2p and Hxt7p [Özcan & Johnston, 1995; Reifenberger et al, 1997; Subtil & Boles, 2012]), or to the contribution of aldose reductases (Richard et al, 1999; Träff et al, 2001, 2002; Ford & Ellis, 2002).

A drop in performance is expected when cells are exposed to perturbations, as they direct nutrients and energy towards maintenance rather than growth or production (Stanley et al, 2010; Vos

et al, 2016). However, our results show that poorly performing strains, such as LBCM1095 and LBCM1079, maintained a robust behavior in the tested perturbation space. We speculate that in strains with robust phenotypes, part of the energy and resources may be directed towards additional regulatory and homeostasis-maintenance pathways even when no environmental perturbations are present, at the expense of lower overall performance.

Two main mechanisms could explain the strong robustness exhibited by LCBM strains and Ethanol Red. The first is bet-hedging, whereby cell-to-cell variation within an isogenic population serves as a survival strategy under different conditions (Bagamery et al, 2020). Strains from the LBCM collection have been isolated from cachaça distilleries around Brazil and resulted from hybridization with *Saccharomyces* strains of varied origin (da Conceição et al, 2015; Araújo et al, 2018; Jacobus et al, 2021). Because of the introduction of new and diversified genetic material, the LBCM populations might have been able to randomly diversify their phenotypes, compared with, for example, CEN.PK113-7D, thereby responding more consistently to environmental perturbations. Bet-hedging implies sometimes a reduced mean fitness (Olofsson et al, 2009), which might explain the low performance of LBCM1079 and LBCM1095. Analysis of the Ethanol Red phylogeny has revealed its close relation to S288C, Y22-3, wine strains, and sugarcane strains, suggesting that its genetic hybridization might also be responsible for its robustness mechanisms (Gronchi et al, 2022). The second mechanism is cross-protection (Dhar et al, 2013), which arises when cells counteract new perturbations based on previous exposure to the same or different perturbations. Even though the robustness of Ethanol Red was slightly lower than that of LBCM1079 and LBCM1095, it nevertheless achieved the highest performance across strains. In Ethanol Red, robustness could also have developed because of cross-resistance mechanisms arising from exposure to different substrates, such as wine, sugarcane or starch.

Earlier investigations have shown that the inoculum might negatively influence the bioprocess outcome mainly because of variations in the physiological and metabolic states of the transferred cells (Keil et al, 2019). We observed a positive correlation between the specific growth rate in the first and second cultivations, and an overall increase of the maximum specific growth rate suggesting that exposing the cells to the same condition before fermentation had a positive effect on their performance, irrespective of perturbation type. Changes in genomic expression patterns triggered by a certain stressor are not required to survive exposure the encountered stressors, but they are fundamental to survive exposure to the same or different stress at a later time (Święciło, 2016). Furthermore, previous evidence has shown that short-term adaptation or acclimatization (van Dijk et al, 2019; Bergen et al, 2022) selects for phenotypes that are more tolerant to previously faced inhibitors (Nielsen et al, 2015). Even if an overall negative correlation was not found between the percentage of improvement of the maximum specific growth rate and the robustness, because of how robustness is defined, we speculate that an increased maximum specific growth rate in the second cultivation might be an indicator of poor robustness of the same phenotype.

The observed significant negative correlations between robustness and performance confirmed the hypothesized trade-offs between these two properties. Whereas others have suggested that

robustness and performance always result in a trade-off (Whitacre, 2012), we found that trade-offs applied only to biomass yield, ethanol yield, and cell dry weight. Instead, specific growth rate exhibited a positive correlation between performance and robustness. This can be explained by specific growth rate having been selected and optimized in many environments through evolution to allow robust-faster-growing cells to dominate the population (Dragosits & Mattanovich, 2013). Natural selection could account for the robust specific growth rate phenotype in our study, and the observed positive correlation between performance and robustness. In contrast, evolutionary mechanisms do not typically favor increased yields, and this lack of optimization may explain the negative correlations we observed between performance and robustness for ethanol and biomass yields and cell dry weight. Similar trade-offs between robustness and performance have been discovered before when measuring ATP and biomass yields (Schuetz et al, 2012). In that study, researchers demonstrated that there is a greater likelihood of survival for cells that prioritize robustness (can switch fast between environments) over maximum performance (highly optimized cells).

Trade-offs appeared in two different forms in our study: first, by way of negative correlations observed between robustness and performance; second, as a reduced performance in some conditions. Yeast cells prioritize different metabolic processes depending on the environment. Because of physical and thermodynamic limitations (Niebel et al, 2019), it is impossible to maximize all metabolic processes simultaneously, resulting in trade-offs.

In conclusion, the design of microbial cell factories would benefit from studies that could reveal trade-off among cellular properties such as in silico objective optimization studies. The methodology and analysis we just presented could help in validation and integration of such simulations. Future research should evaluate more perturbation spaces and different microorganisms, including bacteria and fungi, to determine whether trade-offs follow a more general pattern or should be constrained within singular perturbation spaces and organisms. Finally, studies on evolution of performance and robustness are needed to dig into the possible molecular and genetic markers associated to the two properties, specifically highlighting differences when trade-offs are present.

## Materials and Methods

### Strains

A total of 24 *S. cerevisiae* strains were used. CEN.PK113-7D (Entian & Kötter, 2007) (Scientific Research and Development GmbH) and S288C (University of Milano Bicocca) were two representative laboratory strains. Four industrial strains included: the PE2 WT strain isolated during sugarcane-to-ethanol production (Basso et al, 2008), Ethanol Red (kindly provided by Société Industrielle Lesaffre, Division Leaf), Thermosacc (Lallemand Ethanol Technology), and Red Star (Red Star Yeast, Lesaffre). A set of strains from the LBCM collection (LBCM, Universidade Federal de Ouro Preto, laboratorio de biologia cellular e molecular) (De Araújo Vicente et al, 2006; Araújo et al, 2018) sampled from cachaça production sites in

Brazil were selected: LBCM1001, LBCM1003, LBCM1008, LBCM1013, LBCM1014, LBCM1017, LBCM1030, LBCM1046, LBCM1079, LBCM1095, LBCM1099, LBCM1106, LBCM37, LBCM67, LBCM97, LBCM103, LBCM109, and LBCM110. Cachaça strains were chosen because of their ability to tolerate high concentration of ethanol and perturbations that resemble the one found in second generation biomass fermentation.

## Performance measurement

Performance and robustness measurements were carried out as detailed in our previous study (Trivellin et al, 2022). In short, the yeast strains were grown in Delft chemically defined medium (Bruinenberg et al, 1983; Verduyn et al, 1992) containing 5 g/liter $(NH_4)_2SO_4$, 3 g/liter $KH_2PO_4$, 1 g/liter $MgSO_4 \cdot 7H_2O$, 1 ml trace mineral solution (per L of medium), and 1 ml vitamin solution (per L of medium) (Trivellin et al, 2022). The medium was adjusted to pH 5 with KOH and buffered with 250 mM potassium hydrogen phthalate. Acids, sugars, aldehydes, NaCl, and ethanol were added to the Delft medium to simulate perturbations common during lignocellulosic bioethanol production (Table S1) (Olsson & Hahn-Hägerdal, 1996; Palmqvist et al, 1998; Hohmann, 2002; Koppram et al, 2012; Cavka & Jönsson, 2013; Favaro et al, 2013; Kim, 2018; van Dijk et al, 2019). Briefly, 10 μl glycerol stock containing the *S. cerevisiae* strains were inoculated in 5 ml Delft medium. Precultures were grown overnight at 30°C on a orbital shaker (orbit 1.9 cm) at 250 rpm speed. Screenings were carried out for 48 h, at 30°C, 250 rpm/50 mm orbital shaking in a growth profiler (960; Enzyscreen) using 96-well microtiter plates (CR1496dg) covered with a $CO_2$-release cover (CR1296t), and with a starting $OD_{600}$ of 0.02 in 250 μl.

The final $OD_{600}$ of the cultivation was measured in a plate reader (SPECTROstar nano; BMG LABTECH). The OD values were related to the cell dry weight through a calibration curve previously built for each specific strain in Delft medium (Trivellin et al, 2022). After 48 h of cultivation, the culture broths were filtered, and ethanol and sugar concentrations were determined by enzymatic assays (K-ETOH Ethanol Assay Kit, K-GLUHK-220A D-Glucose HK Assay Kit, K-MANGL D-Mannose/D-Fructose/D-Glucose Assay kit, K-XYLOSE D-Xylose Assay Kit, and K-ARGA L-Arabinose/D-Galactose Assay Kit; Megazyme). Sugar and ethanol concentrations were used to calculate ethanol and biomass yields (Trivellin et al, 2022). Data from the growth profiler (green values) were used to calculate the maximum specific growth rate and length of the lag phase (see Data Availability).

## Robustness quantification

Robustness of each strain (*S*) for a defined phenotype (*i*) across a set of perturbations (*P*) was calculated with the following equation:

$$R_{S,i,P} = -\frac{Fano\,factor}{mean} = -\frac{\sigma^2}{\overline{x}} \cdot \frac{1}{m} \qquad (1)$$

The Fano Factor (variance ($\sigma^2$) divided by the mean ($\overline{x}$) across perturbations) was normalized to the mean of the phenotype across all strains (*m*).

## Investigation of the culture transfer on robustness and performance

To assess the impact of the culture transfer, at the end of the first screening (48 h), 5 μl of culture were reinoculated in 250 μl fresh medium as described above. The plates used in the second screening were incubated for another 48 h under the same conditions as during the first screening. At the end of the second screening, green values were used to calculate the maximum specific growth rate of the second cultivation. The percentage of improvement in maximum specific growth rate performance was measured with the following equation (*μmax1*: maximum specific growth rate of the first cultivation; *μmax2*: maximum specific growth rate of the second cultivation).

$$\%P\mu max = \frac{\mu_{max2} - \mu_{max1}}{\mu_{max1}} \cdot 100 \qquad (2)$$

When *μmax1* was zero and *μmax2* was a finite value, the ratio resulted in an infinite value, which created problems regarding the interpretation of the results. Therefore, infinite values were set to the maximum finite improvement value calculated from the dataset. Spearman correlation tests were performed between the robustness values of all the strains for each phenotype and the mean %P of each strain (scripts with in-line description are available on GitHub [See Data Availability]).

## Data analysis

Dataset containing raw values was preprocessed before performance and robustness analysis. The dataset was trimmed by taking in consideration theoretical values of the yields and experimental errors. The upper boundary for the yields was set at the maximum theoretical ethanol yield on glucose (0.51 + 0.1 g/g), whereas the maximum biomass concentration (39.6 g/liter) was calculated based on the maximum biomass yield measured in chemostat cultivations (0.1 1/h dilution rate [Verduyn, 1991]) and maximum substrate used (65 g/liter glucose). A total of 145 values were excluded from the analysis (out of 10,440) (Fig S5). The Shapiro normality test revealed that the phenotypes did not follow a normal distribution ($P < 1 \times 10^{-38}$); instead, distribution was primarily skewed and multimodal (mostly bimodal) (Figs S6, S7, S8, S9, and S10). The quantile method (percentiles of 0.1% and 99.9%) was used to identify outliers on the trimmed dataset. Thirteen outliers were identified across all the variables corresponding to seven strains (six LBCM strains) and seven perturbations (Fig S11). The outliers were not removed from the dataset. Ethanol yields in the presence of ethanol as a perturbation were excluded from the analysis as we could not differentiate between its consumption and production during cultivation. Spearman's rank-order correlations were performed on the data and associated with their statistical significance (*P*-value). The linear regression lines were included in the graphs mainly to visualize the direction and strenght of the relationships. The standard error of the correlation based on sample size was calculated for performance versus robustness correlations (Bonett & Wright, 2000). Statistical tests and plots were generated in R and

scripts with line-by-line descriptions are available from GitHub (See Data Availability).

## Data Availability

The datasets and R code produced in this study are available on GitHub (https://github.com/cectri/Robustness_Trade-offs/tree/main). The repository contains: raw and processed data, scripts to generate phenotypic variables and preprocessing, exploratory analysis, robustness calculations, correlations, and percentage of improvement scripts. Scripts were generated using previously published R packages (Soetaert et al, 2010; Hall et al, 2014; Giner & Smyth, 2016; Wickham, 2016; McInnes et al, 2018; Bolar, 2019; Neuwirth, 2022; Schloerke et al, 2022; Wilke, 2022; Yutani, 2022; Erik Clarke, 2023; Kassambara, 2023a, 2023b; R Core Team, 2023; Tierney & Cook, 2023; Wickham et al, 2023; Xiao, 2023).

## Supplementary Information

## Acknowledgements

Financial support by Novo Nordisk Foundation grant DISTINGUISHED IN-VESTIGATOR 2019—Research within biotechnology-based synthesis and production (**#0055044**) is gratefully acknowledged. Société Industrielle Lesaffre, Division Leaf, is kindly acknowledged for providing the Ethanol Red strain.

### Author Contributions

C Trivellin: conceptualization, data curation, software, formal analysis, validation, investigation, visualization, methodology, and writing—original draft, review, and editing.
P Rugbjerg: conceptualization, supervision, validation, methodology, project administration, and writing—review and editing.
L Olsson: conceptualization, resources, supervision, funding acquisition, methodology, project administration, and writing—review and editing.

### Conflict of Interest Statement

The authors declare that they have no conflict of interest.

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
