## [Reviewer comments · Life Science Alliance]

Life Science Alliance

Performance and robustness analysis reveals phenotypic trade-offs in yeast

Cecilia Trivellin, Peter Rugbjerg, and Lisbeth Olsson

DOI: <https://doi.org/10.26508/lsa.202302215>

Corresponding author(s): *Lisbeth Olsson, Chalmers University of Technology*

Review Timeline:

Submission Date:	2023-06-13
Editorial Decision:	2023-08-28
Revision Received:	2023-09-25
Editorial Decision:	2023-09-26
Revision Received:	2023-10-20
Accepted:	2023-10-20

Transaction Report:

August 28, 2023

Re: Life Science Alliance manuscript #LSA-2023-02215-T

Prof Lisbeth Olsson
Chalmers University of Technology
Department of Chemical and Biological Engineering
Kemivägen 10
Gothenburg 412 96
Sweden

Dear Dr. Olsson,

Thank you for submitting your manuscript entitled "Performance and robustness analysis reveals phenotypic trade-offs in yeast" to Life Science Alliance. The manuscript was assessed by expert reviewers, whose comments are appended to this letter. We invite you to submit a revised manuscript addressing the Reviewer comments.

Thank you for this interesting contribution to Life Science Alliance. We are looking forward to receiving your revised manuscript.

Sincerely,

B. MANUSCRIPT ORGANIZATION AND FORMATTING:

Reviewer #1 (Comments to the Authors (Required)):

The paper has focus on performance and robustness analysis in yeast strains for lignocellulosic bioethanol production. The work is based on data obtained from 24 different strains that have been cultivated under conditions that simulate specific perturbations. The authors present a method to compare robustness of the strains to cope with these perturbations, and demonstrate different robustness of individual strains. Performance of the strains for robustness analysis is quantified in terms of ethanol yield, biomass yield, growth rate etc. Finally, it is also demonstrated how propagation influences robustness, i.e. cells that have been exposed to specific perturbations in the pre-culture often perform better in the subsequent main cultivation when exposed to the same perturbations.

This is an excellent paper. It is well-written, easy to follow, and contains a detailed interpretation of an extensive dataset that supports the conclusions. The work is relevant for a relatively broad audience.

Reviewer #2 (Comments to the Authors (Required)):

The authors aim investigate the robustness of possible production libraries of strains of different phenotypes in industrially relevant settings. Therefore, they performed high throughput experiments in 96 well plates with different additives which imitate stress conditions in industrial processes. After batch experiments the cell density, recalculated as cell dry weight, length of lag phase, the maximum specific growth rate, and the yields of ethanol and biomass were calculated and compared. Acids, sugars, aldehydes, NaCl, and ethanol were added to the growth medium to simulate stress conditions. The evaluation of the above mentioned data allowed to compare the clones in relation to their robustness to these stressors.

The whole story is well performed with excellent figures and also well described. My only concern is that the authors speak about perturbations. For me a perturbation is a kind of oscillatory stress event. Therefore, I expected in the paper to see different kind of experiments. To my opinion the authors may like to change their description in a way to speak about industrially relevant stress conditions, but not about perturbations.

In conclusion I consider the paper as highly relevant for early process development and the example process focussing on lignocellulose and ethanol producing yeasts is very relevant. The beauty of the approach is that it easily can be applied to other processes also.

Reviewer #3 (Comments to the Authors (Required)):

This work screens different *S. cerevisiae* strains in microtiter plates, under different conditions, relevant in the context of lignocellulosic ethanol production. The authors name the whole set of conditions as the "perturbation space". From growth data obtained in a GrowthProfiler equipment, they calculate some key performance parameters and also a robustness index. The results point to a trade-off between performance and robustness, which had been proposed in previous works from others, in different contexts.

The authors have a previous work on the topic (ACS Synthetic Biology, 2022). It would be important to mention how the present work differs from the previous one, since it really helps the reader to better understand the context.

Line 13: I would avoid the term "bioethanol fermentation", since there is a metabolic process named ethanol fermentation, which oxidises (and does not produce) ethanol. Maybe use "ethanolic or alcoholic fermentation" or "bioethanol production"? Makes it clearer, at least to me.

Lines 27 and 28: I am not sure whether this statement holds true in general. In the biopharma industry, for instance, it is utmost importance that the bioprocess is reproducible and constant. Any changes to the process require certification.

Line 55: I wonder whether "variation of cellular function" is indeed the best termo to be used here. Wouldn't it rather be something like "variation of process performance"? Parameters such as yield, titer and rates are the result of the microbial

population and the environmental conditions, and not solely cellular function.

Line 63: "may be offset by lower specific growth rates" is not clear to me

Lines 67 to 69: "Experimental trade-off validation could explain why strains optimized for maximum performance are less capable of coping with environmental stresses and fluctuations (i.e., present lower robustness)." -> If this is the case, what would happen with those ideas around a "minimal cell"? It seems that they would present decreased robustness, which goes against some proposals that we should use these Taylor-made minimal cells in bioprocesses. Just a thought.

Lines 106 and 107: "phenotypes were measured" -> phenotypes are not directly measured, but rather calculated (even DCW is calculated). Thus, I would rephrase that.

Lines 109 to 111: I think this affirmation requires some clarification. Was it expected that industrial strains would perform better than lab strains in all 5 performance parameters? Aren't biomass yield and ethanol yield somehow inversely correlated, i.e. when one increases the other one decreases?

Lines 115 and 116: "While several studies have pointed to the exceptional performance of PE2 (Soares-Costa et al, 2014; Silva et al, 2018)". ->. what exactly is meant with exceptional performance here? In terms of which of the 5 parameters (or maybe another parameter)? Under which conditions? Performance in one condition does not mean performance in another condition. See e.g. Raghavendran et al (2017) Antonie van Leeuwenhoek: PE-2 does not necessarily lead to higher ethanol titers under cultivation conditions that resemble those found in the sugarcane biorefinery. What really makes it persist in this environment is its capacity to maintain high viability.

Lines 143 to 145: why is that? The authors could at least speculate. Put it simple, $M_{max} = Y_{x/s} \cdot M_{iS}$, which means that if M_{max} is the same and $Y_{x/s}$ was lowered, M_{iS} should increase?

Lines 148 to 150: I think the reference Groter de Vries indicated is not the work that suggests that *L. Amylovorus* might lead to a 3% increase in the ethanol yield. This work is probably Lino et al (2021) Nature Communications?

Lines 151 to 156: is it expected that pentoses should lead to higher ethanol yields? If not, what is the reason for these observations?

Lines 266 and 267: would be interesting to switch perturbations, to see how that affects the growth rate.

Line 424: "To encompass the natural diversity of wild-type strains" -> Only cachaça strains were used, thus this not encompass the natural diversity of WT strains. I would rephrase it.

Line 425: Clarify what the acronym LBCM means.

Line 433: Sure that the correct reference for the Delft medium is "Bruinenberg et al, 1983" and not Verduyn et al, 1992?

Line 426: "250 mM potassium hydrogen phthalate." -> seems like a high concentration of buffer. Are there any data of previous information showing this does not influence yeast performance?

Lines 446 and 447: "The OD values were related to the cell dry weight through a calibration curve (Trivellin et al, 2022)" -> please indicate how this calibration was performed and for which strain. Was there one curve for each strain? Or was one curve obtained for one strain and extrapolated for the others? If yes, how far is this valid?

Lines 452 and 453: "Sugar and ethanol concentrations were used to calculate ethanol and biomass yields." -> Could the authors show the formula? How were initial values measured or considered? Is it sure that ethanol was not consumed or evaporated in the experiments?

Lines 482 and 483: "the maximum biomass concentration (39.6 g/L) was calculated based on the theoretical maximum biomass yield and maximum substrate used (65 g/L glucose)" -> I could not understand how the authors reached this 39.6 value, could this be clarified?

In this work, single perturbations were used. The authors could comment on what would happen if combinations of these perturbations were used. Should we expect a simple addition of effects of each perturbation or is it foreseen that synergies will exist, in such a way that the responses will be different from a single sum of effects? Also, in a real industrial process, there is a perturbation dynamics, how would that affect microbial performance and robustness?

Finally, maybe the editor (or journal) would like to ask a statistician to revise the work, since many different statistical approaches were applied and I am not able not evaluate them.

Manuscript ID: Life Science Alliance manuscript #LSA-2023-02215-T

Authors: Trivellin Cecilia; Rugbjerg Peter; Olsson Lisbeth

Response to Reviewers

We thank the editor for giving us the opportunity to submit a revised draft of our manuscript. We appreciate the effort the editor and the reviewers dedicated to providing valuable comments to our manuscript. We have carefully revised our manuscript according to the comments and we hope that it has significantly improved.

Point-by-point responses to the reviewer's comments are highlighted in red, please see below.

Reviewer(s)' Comments to Author:

Reviewer #1:

The paper has focus on performance and robustness analysis in yeast strains for lignocellulosic bioethanol production. The work is based on data obtained from 24 different strains that have been cultivated under conditions that simulate specific perturbations. The authors present a method to compare robustness of the strains to cope with these perturbations, and demonstrate different robustness of individual strains. Performance of the strains for robustness analysis is quantified in terms of ethanol yield, biomass yield, growth rate etc. Finally, it is also demonstrated how propagation influences robustness, i.e. cells that have been exposed to specific perturbations in the pre-culture often perform better in the subsequent main cultivation when exposed to the same perturbations.

This is an excellent paper. It is well-written, easy to follow, and contains a detailed interpretation of an extensive dataset that supports the conclusions. The work is relevant for a relatively broad audience.

Response: We thank the reviewer for the kind comment.

Reviewer #2:

The authors aim investigate the robustness of possible production libraries of strains of different phenotypes in industrially relevant settings. Therefore, they performed high throughput experiments in 96 well plates with different additives which imitate stress conditions in industrial processes. After batch experiments the cell density, recalculated as cell dry weight, length of lag phase, the maximum specific growth rate, and the yields of ethanol and biomass were calculated and compared. Acids, sugars, aldehydes, NaCl, and ethanol were added to the growth medium to simulate stress conditions. The evaluation of the above mentioned data allowed to compare the clones in relation to their robustness to these stressors.

The whole story is well performed with excellent figures and also well described. My only concern is that the authors speak about perturbations. For me a perturbation is a kind of oscillatory stress event. Therefore, I expected in the paper to see different kind of experiments. To my opinion the authors may like to change their description in a way to speak about industrially relevant stress conditions, but not about perturbations.

In conclusion I consider the paper as highly relevant for early process development and the example process focussing on lignocellulose and ethanol producing yeasts is very relevant. The beauty of the approach is that it easily can be applied to other processes also.

Response: We appreciate the reviewer's suggestion regarding the terminology related to perturbations. To maintain consistency with our prior publications and the literature we have cited, we have opted to retain the term "perturbation" when discussing general stresses within an environment. We have added a sentence to clarify that variable conditions and perturbations can be used interchangeably (line 34: "*The variable conditions, also referred as perturbations,...*"). However, we acknowledge that it might create confusion when referring to the experimental singular perturbations in our study. Therefore, we have made the necessary revisions, replacing perturbations with the term "condition" across the whole document.

Reviewer #3:

This work screens different *S. cerevisiae* strains in microtiter plates, under different conditions, relevant in the context of lignocellulosic ethanol production. The authors name the whole set of conditions as the "perturbation space". From growth data obtained in a GrowthProfiler equipment, they calculate some key performance parameters and also a robustness index. The results point to a trade-off between performance and robustness, which had been proposed in previous works from others, in different contexts.

The authors have a previous work on the topic (ACS Synthetic Biology, 2022). It would be important to mention how the present work differs from the previous one, since it really helps the reader to better understand the context.

Response: We appreciate the reviewer's feedback, emphasizing the need to differentiate the current study from our previous work. We have addressed this concern by changing the last paragraph of the introduction with a concise explanation of how this study builds upon our earlier research (lines 122-125):

“While our earlier work primarily focused on the development and validation of the robustness quantification method itself, in the current study our aim is to demonstrate its practical application. Specifically, our study...”

Line 13: I would avoid the term "bioethanol fermentation", since there is a metabolic process named ethanol fermentation, which oxidises (and does not produce) ethanol. Maybe use "ethanolic or alcoholic fermentation" or "bioethanol production"? Makes it clearer, at least to me.

Response: We agree with the reviewer’s point. The term has been changed to bioethanol production.

Lines 27 and 28: I am not sure whether this statement holds true in general. In the biopharma industry, for instance, it is of utmost importance that the bioprocess is reproducible and constant. Any changes to the process require certification.

Response: The reviewer raised a valid point regarding the stringent requirements for process stability (especially relevant in the biopharma industry). While economic feasibility and high yields are significant considerations, they are not the sole factors in bioprocess evaluation. Reproducibility and product quality hold equal importance. Consequently, emphasizing robustness is paramount, ensuring that stochastic perturbations do not compromise process outcomes. We have revised lines 27-29 to reflect this crucial aspect:

“To achieve high yields while ensuring reproducibility and high-quality in bioprocesses, it is essential to address the factors contributing to process variability.”

Line 55: I wonder whether "variation of cellular function" is indeed the best term to be used here. Wouldn't it rather be something like "variation of process performance"? Parameters such as yield, titer and rates are the result of the

microbial population and the environmental conditions, and not solely cellular function.

Response: In our previous work we denoted cellular function (phenotype) as the attribute of the system under assessment for its performance and robustness. In this instance, we concur with the reviewer that the wording within this context might lead to ambiguity. Consequently, we have replaced "variation of cellular function" with "variation of the performance" in line 75.

Line 63: "may be offset by lower specific growth rates" is not clear to me

Response: We have rephrased the examples in lines 80 to 86:

"Microorganisms appear to exhibit trade-offs between performance and robustness of one or more phenotypes (Kitano, 2007, 2010), which could be important to improving strain and process development. For example, in bacteria, antibiotic resistance and extracellular enzyme production have been shown to trade off with the specific growth rate and a similar trade-off has been observed between melanin production and specific growth rate in fungi (Andersson, 2006; Lovero & Treseder, 2021; Ramin & Allison, 2019)."

Lines 67 to 69: "Experimental trade-off validation could explain why strains optimized for maximum performance are less capable of coping with environmental stresses and fluctuations (i.e., present lower robustness)." -> If this is the case, what would happen with those ideas around a "minimal cell"? It seems that they would present decreased robustness, which goes against some proposals that we should use these Taylor-made minimal cells in bioprocesses. Just a thought.

Response: We thank the reviewer for this insightful perspective. The prospect of incorporating a minimal cell into a bioprocess holds significant promise. It has the potential to address a multitude of challenges currently encountered in

bioprocessing: declining productivity resulting from the heterogenous cell populations, persistent issues like plasmid loss and genetic instability, to name a few. Essentially, a minimal cell embodies a set of indispensable genes working collaboratively towards a specific task within a precisely defined environment. This description characterizes a cell factory (the minimal cell) whose metabolic activity is well-understood, and any unanticipated metabolic fluctuations approach negligible levels. Significant concerns have been reported regarding the use of minimal cells, especially yeast, where cellular complexity poses additional challenges (Ziegler & Takors, 2019). Firstly, eliminating entire operons or pathways, seemingly redundant for a given task, may inadvertently remove vital transcription factors. Secondly, while single nucleotide deletions might be inconsequential, substantial genome deletions could disrupt chromosomal organization and structure, critical for regulating protein expression levels (Carpentier *et al*, 2005). The mechanisms of robustness are yet to be fully understood; however, they hold major importance in the regulation of cell processes against variation of any sort. For instance, the significance of gene or pathway redundancy in ensuring robustness has been well-documented (Olsson *et al*, 2022), something that would lack in a minimal cell. In summary, while the concept of a minimal cell factory is undoubtedly appealing, our assumption is that a minimal cell may exhibit limited adaptability in processes marked by variations, whether triggered by perturbations, contamination, genetic mutations, or alterations in substrate composition. This potential limitation could arise from the deletion of genes deemed "non-essential" in one stable environment but proving indispensable in others. Based on this stand-point, we have not made any textual changes in the manuscript regarding this point.

Lines 106 and 107: "phenotypes were measured" -> phenotypes are not directly measured, but rather calculated (even DCW is calculated). Thus, I would rephrase that.

Response: We have changed the wording in line 166.

Lines 109 to 111: I think this affirmation requires some clarification. Was it expected that industrial strains would perform better than lab strains in all 5 performance parameters? Aren't biomass yield and ethanol yield somehow inversely correlated, i.e. when one increases the other one decreases?

Response: We agree with the reviewer regarding the clarification of line 167 to 169:

"Although it has been reported that industrial strains grow faster than laboratory strains (Kong et al, 2018; Yi & Alper, 2022) due to their ability to handle perturbations in industrial settings, no significant differences were detected."

Industrial strains are generally better than laboratory strains but for different purposes (Kong et al, 2018; Yi & Alper, 2022). For instance, strains like PE-2, which are employed in ethanol production, are expected to exhibit higher ethanol yield and a greater maximum specific growth rate in environments where they are typically utilized, such as sugarcane-based fermentation. Commercial strains like RedStar®, designed for baking applications, should exhibit high CO₂ production and cell dry weight due to their specific industrial requirements. Maximum specific growth rates are generally higher as the industrial strains are more tolerant to stresses. We agree that a strain with optimized fluxes towards ethanol will trade-off with biomass production due to metabolic constraints. However, our data analysis revealed a significant negative Spearman correlation only between ethanol yield and biomass yield in the presence of hexoses ($R = -0.14$, $p = 0.002$). In other conditions, the correlation was either close to zero or statistically insignificant.

Lines 115 and 116: "While several studies have pointed to the exceptional performance of PE2 (Soares-Costa et al, 2014; Silva et al, 2018)". ->. what exactly is meant with exceptional performance here? In terms of which of the 5 parameters (or maybe another parameter)? Under which conditions? Performance in one condition does not mean performance in another condition. See e.g. Raghavendran et al (2017) Antonie van Leeuwenhoek: PE-2 does not necessarily lead to higher ethanol titers under cultivation conditions that resemble those found in the sugarcane biorefinery. What really makes it persist in this environment is its capacity to maintain high viability.

Response: We thank the reviewer for this comment. In response, we have now included both the parameters assessed and the growth conditions for PE-2 cultivation in the manuscript. Our results do corroborate with the findings of Raghavendran et al. (2017), who emphasized that strain viability is crucial for consistent performance (in terms of ethanol titers) of PE-2 yeast during sugarcane ethanol fermentation. We have also included additional studies (Araújo *et al*, 2018; Pereira *et al*, 2014), which attribute PE-2's fermentation capacity to its ability to degrade toxic compounds (e.g., furfural) and to its elevated intracellular trehalose content. These valuable insights and their respective references have been integrated into the manuscript, spanning from line 172 to 192:

"PE-2 strain has been reported to have a high CO₂ production, final ethanol production (close to 95% of the theoretical yield in E. globulus wood (EGW) hydrothermal hydrolysate anaerobic fermentation) fast sugar utilization and faster degradation of furfural and 5-hydroxymethyl-2-furaldehyde (Soares-Costa et al, 2014; Pereira et al, 2014, 2010). Furthermore, PE-2 has also been shown to dominate and persist in Brazilian distilleries probably due to its high viability and high

specific growth rate (Araújo et al, 2018; Raghavendran et al, 2017). In our study, the specific growth rate of PE2 was 20% lower than the average across all strains”.

Lines 143 to 145: why is that? The authors could at least speculate. Put it simple, $M_{max} = Y_{x/s} \cdot \mu_{max}$, which means that if M_{max} is the same and $Y_{x/s}$ was lowered, μ_{max} should increase?

Response: Considering fermentation kinetics, it is plausible that a reduction in $Y_{x/s}$, while keeping μ_{max} constant, would lead to an increased specific substrate consumption rate. We have included this speculation into the manuscript from line 231 to 238:

“Considering fermentation kinetics, a decreased yield of biomass on substrate while keeping the specific growth rate constant may suggest an increase in the specific rate of substrate consumption. Weak acids are not inhibitory enough to slow down the anabolism, therefore growth can proceed at high specific rate. However, ATP is required for counteracting the effects of the acidification, which diverts the carbon source from anabolism to catabolism. This decreases the yield and increases the specific substrate consumption rate, and the specific production rate of energy-related primary metabolites.”

We have removed the explanation of this phenomena in the discussion as it was a repetition of the above.

Lines 148 to 150: I think the reference Groter de Vries indicated is not the work that suggests that *L. Amylovorus* might lead to a 3% increase in the ethanol yield. This work is probably Lino et al (2021) Nature Communications?

Response: We appreciate the reviewer for identifying this error, which has been successfully rectified. Additionally, during this correction process, we've recognized

that the pKa of lactic acid is 3.79, and at pH 5, the acid predominantly exists in its dissociated form. Consequently, this makes it exceptionally challenging for the acid to penetrate the yeast cells and affecting their metabolic processes. We have added this explanation from line 240 to 242:

“This was likely because the pKa of lactic acid is 3.79, and at a pH of 5, the majority of the acid existed in its dissociated form, making it less likely to penetrate the cells.”.

Lines 151 to 156: is it expected that pentoses should lead to higher ethanol yields? If not, what is the reason for these observations?

Response: We acknowledge that previous research did not provide a satisfactory explanation for the phenomenon we observed. In our discussion, we have hypothesized on why this phenomena was observed (lines 473 to 476: *“We hypothesize that this might be attributed to an increased efficiency of some hexose transporters (specifically Hxt2p and Hxt7p (Özcan & Johnston, 1995; Subtil & Boles, 2012; Reifenberger et al, 1997)), or to the contribution of aldose reductases (Träff et al, 2001, 2002; Ford & Ellis, 2002; Richard et al, 1999)).* However, it is essential to highlight that further experiments conducted on a larger scale are necessary to delve deeper into this aspect. Given that our study was conducted as a comparative study in a high-throughput fashion, drawing conclusive insights into the underlying physiology is challenging without the measurement of rates and concentrations of metabolites like glycerol, acetate, or CO₂. Nevertheless, our study offers valuable initial observations of an intriguing effect that demands further investigation.

Lines 266 and 267: would be interesting to switch perturbations, to see how that affects the growth rate.

Response: We think the reviewer might refer to line 276 and 277 here, commenting on cell transfer to a different medium. This indeed presents an intriguing point.

Numerous research studies have explored the mechanisms of cross-stress resistance, which are currently understood to be influenced by the specific characteristics of the initial stress rather than being governed by universal stress response mechanisms. No further comments were put into the text on this aspect.

Line 424: "To encompass the natural diversity of wild-type strains" -> Only cachaça strains were used, thus this not encompass the natural diversity of WT strains. I would rephrase it.

Response: We agree with the reviewer's point. We have rephrased line 589 to 591: *"Cachaça strains were chosen because of their ability to tolerate high concentration of ethanol as well as perturbations that resemble the one found in second generation biomass fermentation."*

Line 425: Clarify what the acronym LBCM means.

Response: We have referenced the paper where the strains were isolated and specified the lab and university (line 584 to 586). LBCM is just the acronym of the laboratory which isolated the strains: laboratório de biologia celular e molecular (Universidade Federal de Ouro Preto).

Line 433: Sure that the correct reference for the Delft medium is "Bruinenberg et al, 1983" and not Verduyn et al, 1992?

Response: Both references contain the same minimal medium composition, so we preferred to cite to the study that published this first. The only difference is that in Bruinenberg et al, 1983 they gave a list of suggestions to use instead of ammonium sulfate. We have added the Verduyn et al, 1992 reference as well in case the community is more familiar to that reference.

Line 426: "250 mM potassium hydrogen phthalate. " -> seems like a high concentration of buffer. Are there any data of previous information showing this does not influence yeast performance?

Response: We acknowledge that the buffer concentration of 250 mM in our study is notably higher than the 100 mM or 50 mM concentrations typically employed in prior research. We considered that since all the strains were cultivated using the same buffer, it would still be valid to compare their performance and robustness. Moreover, the potential increase in osmotic pressure due to the higher buffer concentration could be viewed as an additional perturbation akin to industrial conditions. We conducted a test to examine the impact of buffer concentration on our cultivation process. This experiment was carried out under anaerobic conditions using the CEN.PK113-7D strain in 50 mL shake flasks. We utilized 5 mL Delft media (containing 2% glucose) supplemented with varying buffer concentrations: 0 mM, 50 mM, 100 mM, 150 mM, 200 mM, 250 mM, 300 mM, and 350 mM. After a 24-hour incubation period, we did not observe significant differences in the final OD600 of the strain across these buffer concentrations. However, we did measure cell volume and cell diameter and noted a 30% reduction in cell volume and a 10% reduction in cell diameter for strains grown in 250 mM buffer compared to those in the absence of buffer (0 mM).

Lines 446 and 447: "The OD values were related to the cell dry weight through a calibration curve (Trivellin et al, 2022)" -> please indicate how this calibration was performed and for which strain. Was there one curve for each strain? Or was one curve obtained for one strain and extrapolated for the others? If yes, how far is this valid?

Response: We have provided a concise overview of the calibration curve construction process, without delving into extensive repetition since we have previously detailed this procedure in our earlier publication (line 616-617). A separate calibration curve was established for each strain, and the slope specific to each strain was used to convert OD600 values into cell dry weight (CDW).

Lines 452 and 453: "Sugar and ethanol concentrations were used to calculate ethanol and biomass yields." -> Could the authors show the formula? How were initial values measured or considered? Is it sure that ethanol was not consumed or evaporated in the experiments?

Response: The formulas employed for yield calculations have been previously documented in our published work, with a reference now included to clarify this in the methods section. For the determination of ethanol and residual sugars, we conducted measurements using Megazyme kits, which rely on NADH absorbance. Ethanol yield was derived by dividing the final ethanol concentration by the total sugars consumed, while biomass yield was calculated as the final biomass divided by the total sugars consumed. In our cultivation in the growth profiler, we utilized a CO₂ release cover (CR1296t) designed to allow the exit of CO₂ while minimizing passive diffusion. This cover is particularly suitable for anaerobic cultivations. Notably, we did not observe a diauxic shift in any of the growth curves, suggesting that ethanol was not consumed after the fermentation phase. While some ethanol evaporation is common in shake flasks, we speculate that its impact was minimal due to the use of the anaerobic cover. However, we want to emphasize that if such evaporation occurred, it was consistent across all cultivations, ensuring the comparability of the collected data. Furthermore, it's important to clarify that this study is primarily intended for inter-strain comparisons and trend analysis rather than

exact value comparisons. We are aware that results may differ in scale-up and higher-volume fermentations.

Lines 482 and 483: "the maximum biomass concentration (39.6 g/L) was calculated based on the theoretical maximum biomass yield and maximum substrate used (65 g/L glucose)" -> I could not understand how the authors reached this 39.6 value, could this be clarified?

Response: We thank the reviewer for this comment. This particular step was implemented primarily as a filter to eliminate biologically implausible data points that may have arisen during our data collection process. To estimate the maximum biomass concentration, we utilized the biomass yield on glucose of *Saccharomyces cerevisiae* chemostat cultivation conducted at a 0.1 (1/h) dilution rate, which was 0.51 (g/g) (Verduyn, 1991), plus an experimental error of ± 0.1 g/g. Therefore, we used the maximum yield reported in previous studies, including the experimental error margin leading us to a maximum yield of 0.61 g/g. Assuming a glucose consumption of 65 g/L (the maximum substrate concentration in our experiment), this calculation led to a maximum biomass concentration of 39.6 g/L (maximum potential experimental yield = $0.61 \text{ g/g} * 65 \text{ g/L glucose consumed}$). The above is reported in the text (line 658 to 662):

"The upper boundary for the yields was set at the maximum theoretical ethanol yield on glucose (0.51 + 0.1 g/g), while the maximum biomass concentration (39.6 g/L) was calculated based on the maximum biomass yield measured in chemostat cultivations (0.1 1/h dilution rate (Verduyn, 1991)) and maximum substrate used (65 g/L glucose)."

In this work, single perturbations were used. The authors could comment on what would happen if combinations of these perturbations were used. Should we expect a

simple addition of effects of each perturbation or is it foreseen that synergies will exist, in such a way that the responses will be different from a single sum of effects? Also, in a real industrial process, there is a perturbation dynamics, how would that affect microbial performance and robustness?

Response: The reviewer raised an interesting point. Previous literature reported on the negative impact of synergistic effects on the performance (Vanmarcke *et al*, 2021; Liu *et al*, 2004; Peetermans *et al*, 2021). When calculating robustness, we think that if single conditions are tested with the method presented in this study (enough to cover a statistically significant perturbation space), the differences in performance should be already depicted by the robustness score, making the contribution of the synergistic effects and the perturbations dynamics insignificant to the overall robustness score. In addition, it was outside the scope of the present study to investigate more complicated perturbation spaces such that precisely mimic synergistic effect or population dynamics. The generality of our approach would allow such approaches in future studies.

Finally, maybe the editor (or journal) would like to ask a statistician to revise the work, since many different statistical approaches were applied, and I am not able not evaluate them.

Response: we would like to provide some additional information on the statistical tests used in our study to facilitate the evaluation of the data analysis.

Performance calculations:

- The function `all_splines` parameter `r2` was used to filter out the strains that did not grow. If the model did not fit the data points (usually because strains did not grow or if data were noisy) that the function returned a low `r2` (we used 0.985 as a cut). The `mumax` was calculated automatically by the function

based on the inflection of the slope of the growth curve in the exponential phase.

Data preprocessing:

- outlier detection: we used a quintile method (quantile function in R, stats library) with lower and upper bound corresponding to 0.001 and 0.999. However, we did not filter out the outliers as they were still within the reasonable biological boundaries.
- The shapiro.test function from the ggpubr library was used to check whether the distribution of the tested parameters was normal. A p-value < 0.05 indicates that the test is significantly different from a normal distribution (null-hypothesis = sample distribution is normal). Normality test can be sensitive to small sample sizes, but this does not apply to our study as we have a large sample size.

Exploratory analysis:

- To evaluate if the difference in performance among different groups of conditions were significant, we used the function compare_means also from the ggpubr package in R. We choose the Wilcox.test as a method. We did not choose a normal t-test or ANOVA test as we previously checked the normality of our data and the distributions were not-normal. The Wilcoxon rank sum test is a non-parametric alternative to use with non-normally distributed data.
- To check correlations of performance among different parameters (phenotypes) we used a Spearman correlation test. The correlations were calculated using the function ggscatter from the ggpubr library. Same as above the Spearman correlation test was used among others (for example Pearson correlation test) to check whether two parameters are related by a

monotonic relationship (not linear like in Pearson). Spearman is a rank correlation method that is well suited for non-normally distributed data. The p-value <0.05 suggests that the correlation is significant.

Robustness quantification and second cultivation:

- We used the same method as above to check correlations between performance and robustness and between the specific growth rate of the first and second growth (Spearman correlations can also be used with small sample size).
- We used the function `se.spear` within the `vcmeta` library to estimate the error of the spearman correlations based on estimated correlation and sample size.

References:

- Andersson DI (2006) The biological cost of mutational antibiotic resistance: any practical conclusions? *Curr Opin Microbiol* 9: 461–465
- Araújo TM, Souza MT, Diniz RHS, Yamakawa CK, Soares LB, Lenczak JL, de Castro Oliveira JV, Goldman GH, Barbosa EA, Campos ACS, *et al* (2018) Cachaça yeast strains: alternative starters to produce beer and bioethanol. *Antonie van Leeuwenhoek, International Journal of General and Molecular Microbiology* 111: 1749–1766
- Carpentier AS, Torrèsani B, Grossmann A & Hénaut A (2005) Decoding the nucleoid organisation of *Bacillus subtilis* and *Escherichia coli* through gene expression data. *BMC Genomics* 6: 84
- Ford G & Ellis EM (2002) Characterization of Ypr1p from *Saccharomyces cerevisiae* as a 2-methylbutyraldehyde reductase. *Yeast* 19: 1087–1096
- Kitano H (2007) Towards a theory of biological robustness. *Mol Syst Biol* 3: 137

- Kitano H (2010) Violations of robustness trade-offs. *Mol Syst Biol* 6: 384
doi:10.1038/msb.2010.40 [PREPRINT]
- Kong II, Turner TL, Kim H, Kim SR & Jin YS (2018) Phenotypic evaluation and characterization of 21 industrial *Saccharomyces cerevisiae* yeast strains. *FEMS Yeast Res* 18: 1
- Liu ZL, Slininger PJ, Dien BS, Berhow MA, Kurtzman CP & Gorsich SW (2004) Adaptive response of yeasts to furfural and 5-hydroxymethylfurfural and new chemical evidence for HMF conversion to 2,5-bis-hydroxymethylfuran. *J Ind Microbiol Biotechnol* 31: 345–352
- Lovero KG & Treseder KK (2021) Trade-Offs Between Growth Rate and Other Fungal Traits. *Frontiers in Forests and Global Change* 4: 197
- Olsson L, Rugbjerg P, Torello Pianale L & Trivellin C (2022) Robustness: linking strain design to viable bioprocesses. *Trends Biotechnol*
- Özcan S & Johnston M (1995) Three Different Regulatory Mechanisms Enable Yeast Hexose Transporter (HXT) Genes To Be Induced by Different Levels of Glucose. *Mol Cell Biol* 15: 1564–1572
- Peetermans A, Foulquié-Moreno MR & Thevelein JM (2021) Mechanisms underlying lactic acid tolerance and its influence on lactic acid production in *Saccharomyces cerevisiae*. *Microbial Cell* 8: 111
- Pereira FB, Guimarães PMR, Teixeira JA & Domingues L (2010) Selection of *Saccharomyces cerevisiae* strains for efficient very high gravity bio-ethanol fermentation processes. *Biotechnol Lett* 32: 1655–1661
- Pereira FB, Romaní A, Ruiz HA, Teixeira JA & Domingues L (2014) Industrial robust yeast isolates with great potential for fermentation of lignocellulosic biomass. *Bioresour Technol* 161: 192–199

- Raghavendran V, Basso TP, da Silva JB, Basso LC & Gombert AK (2017) A simple scaled down system to mimic the industrial production of first generation fuel ethanol in Brazil. *Antonie van Leeuwenhoek, International Journal of General and Molecular Microbiology* 110: 971–983
- Ramin KI & Allison SD (2019) Bacterial Tradeoffs in Growth Rate and Extracellular Enzymes. *Front Microbiol* 10: 2956
- Reifenberger E, Boles E & Ciriacy M (1997) Kinetic Characterization of Individual Hexose Transporters of *Saccharomyces Cerevisiae* and their Relation to the Triggering Mechanisms of Glucose Repression. *Eur J Biochem* 245: 324–333
- Richard P, Toivari MH & Penttilä M (1999) Evidence that the gene YLR070c of *Saccharomyces cerevisiae* encodes a xylitol dehydrogenase. *FEBS Lett* 457: 135–138
- Soares-Costa A, Nakayama DG, Andrade L de F, Catelli LF, Bassi APG, Ceccato-Antonini SR & Henrique-Silva F (2014) Industrial PE-2 strain of *Saccharomyces cerevisiae*: from alcoholic fermentation to the production of recombinant proteins. *N Biotechnol* 31: 90–97
- Subtil T & Boles E (2012) Competition between pentoses and glucose during uptake and catabolism in recombinant *Saccharomyces cerevisiae*. *Biotechnol Biofuels* 5: 14
- Träff KL, Cordero RRO, van Zyl WH & Hahn-Hägerdal B (2001) Deletion of the GRE3 Aldose Reductase Gene and Its Influence on Xylose Metabolism in Recombinant Strains of *Saccharomyces cerevisiae* Expressing the xylA and XKS1 Genes. *Appl Environ Microbiol* 67: 5668–5674
- Träff KL, Jönsson LJ & Hahn-Hägerdal B (2002) Putative xylose and arabinose reductases in *Saccharomyces cerevisiae*. *Yeast* 19: 1233–1241

- Vanmarcke G, Demeke MM, Foulquié-Moreno MR & Thevelein JM (2021) Identification of the major fermentation inhibitors of recombinant 2G yeasts in diverse lignocellulose hydrolysates. *Biotechnol Biofuels* 14
- Verduyn C (1991) Physiology of yeasts in relation to biomass yields. *Antonie van Leeuwenhoek* 1991 60:3 60: 325–353
- Yi X & Alper HS (2022) Considering Strain Variation and Non-Type Strains for Yeast Metabolic Engineering Applications. *Life* 12
- Ziegler M & Takors R (2019) Reduced and minimal cell factories in bioprocesses: Towards a streamlined chassis. *Minimal Cells: Design, Construction, Biotechnological Applications*: 1–44

September 26, 2023

RE: Life Science Alliance Manuscript #LSA-2023-02215-TR

Prof. Lisbeth Olsson
Chalmers University of Technology
Department of Chemical and Biological Engineering
Kemivägen 10
Gothenburg 412 96
Sweden

Dear Dr. Olsson,

Thank you for submitting your revised manuscript entitled "Performance and robustness analysis reveals phenotypic trade-offs in yeast". We would be happy to publish your paper in Life Science Alliance pending final revisions necessary to meet our formatting guidelines.

- please add ORCID ID for the corresponding author--you should have received instructions on how to do so
- please add your main, supplementary figure, and table legends to the main manuscript text after the references section
- please upload a clean version of the manuscript file without tracking changes
- please label Supplementary Table as S1, instead of ST1, and correct its call-out
- please incorporate the references from the Supplemental Material file into the Reference list provided in the main manuscript file

A. FINAL FILES:

B. MANUSCRIPT ORGANIZATION AND FORMATTING:

Sincerely,

October 20, 2023

RE: Life Science Alliance Manuscript #LSA-2023-02215-TRR

Prof. Lisbeth Olsson
Chalmers University of Technology
Life Sciences
Gothenburg 412 96
Sweden

Dear Dr. Olsson,

Thank you for submitting your Resource entitled "Performance and robustness analysis reveals phenotypic trade-offs in yeast". It is a pleasure to let you know that your manuscript is now accepted for publication in Life Science Alliance. Congratulations on this interesting work.

DISTRIBUTION OF MATERIALS:

Again, congratulations on a very nice paper. I hope you found the review process to be constructive and are pleased with how the manuscript was handled editorially. We look forward to future exciting submissions from your lab.

Sincerely,
